

**Characterization of black carbon-containing fine particles**
**in Beijing during wintertime**
Junfeng Wang[1], Dantong Liu[2], Xinlei Ge[1]*, Yangzhou Wu[1], Fuzhen Shen[1], Mindong Chen[1], Jian
Zhao[3,4], Conghui Xie[3,4], Qingqing Wang[3], Weiqi Xu[3,4], Jie Zhang[5], Jianlin Hu[1], James Allan[2,6],
Rutambhara Joshi[2], Pingqing Fu[3], Hugh Coe[2] and Yele Sun[3,4]
[1]Jiangsu Key Laboratory of Atmospheric Environment Monitoring and Pollution Control, School of
Environmental Science and Engineering, Nanjing University of Information Science and Technology,
Nanjing 210044, China
[2]School of Earth and Environmental Sciences, University of Manchester, M13 9PL, Manchester, UK
[3]State Key Laboratory of Atmospheric Boundary Layer Physics and Atmospheric Chemistry, Institute
of Atmospheric Physics, Chinese Academy of Sciences, Beijing 100029, China
[4]University of Chinese Academy of Sciences, Beijing 100049, China
[5]Atmospheric Sciences Research Center, University at Albany, State University of New York, NY,
12203,USA
[6]National Centre for Atmospheric Science, University of Manchester, M13 9PL, Manchester, UK
*Corresponding author, Email: caxinra@163.com
Phone: +86-25-58731394



**Abstract**

Refractory black carbon (BC) is a product from incomplete combustion of fossil fuel, biomass and biofuel, etc. By mixing with other species, BC can play significant roles in climate change, visibility impairment and human health. Such BC-containing particles in very densely-populated megacities, like Beijing, may have specific sources and properties, that are very important to the haze formation and air quality. In this work, we characterized exclusively the BC-containing particles only in urban Beijing, by using a laser-only Aerodyne soot particle aerosol mass spectrometer (SP-AMS), as a part of the Air Pollution and Human Health (APHH) 2016 winter campaign. The average mass ratio of coating-to-BC ($R_{BC}$) was found to be ~5.0, much smaller than those of highly aged BC, indicating dominant contributions from primary emissions. Positive matrix factorization indeed shows the dominance of fossil fuel and biomass burning organics. Yet secondary species, including both sulfate, nitrate and oxygenated organic aerosol (OA) species, could have significant impacts on the properties of BC-containing particles, especially for ones with larger BC core sizes and thicker coatings. Analyses of the sources and diurnal cycles of organic coating reveal significant afternoon photochemical production of secondary OA (SOA), as well as the nighttime production of a portion of highly oxygenated OA. Besides SOA, photochemical production of nitrate, not sulfate, was very important. Further investigations on BC-containing particles at different periods show that, on average, more polluted periods would have more contributions from secondary species, and more thickly coated BC tended to associate with more secondary species, indicating the important role of chemical aging to the air pollution in urban Beijing during wintertime. However, for individual pollution events, aqueous-phase production of sulfate, nitrate and SOA might aggravate the pollution obviously under high relative humidity conditions, while sometimes local primary emissions (coal and biomass burning) could lead to serious and extremely polluted event too.



## 1. Introduction

Black carbon (BC) is generated from incomplete combustion of carbon-based fuels (Ramanathan and Carmichael, 2008), and can exert significant impacts on global and regional climate, planetary boundary layer height (PBLH), air quality and human health, etc. (Lee et al., 2017;Bond et al., 2013;Ding et al., 2016). BC can strongly absorb solar radiation and warm up the atmosphere directly. By internally or externally mix with non-BC materials (coatings, including co-emitted primary organics/inorganics and secondary materials that associate with BC) (Chen et al., 2016a;Lee et al., 2017;Wang et al., 2017a), the properties and morphologies of BC might be altered greatly (Liu et al., 2013;Liu et al., 2017;Liu et al., 2015;Cappa et al., 2012;Peng et al., 2016). Thick coating can increase the mass absorption cross section of BC, thus enhance the light absorption of BC core via "lensing effect" (Jacobson, 2011;Liu et al., 2015;Pokhrel et al., 2017). However, coating thickness of BC-containing particles significantly depends on their sources/chemical composition and aging processes, thus there are great uncertainties on light absorption enhancement ($E_{abs}$) of BC as well as its global radiative forcing (Cappa et al., 2012;Liu et al., 2017;Cui et al., 2016;Liu et al., 2015). For instance, the mass ratio of coatings to BC core ($R_{BC}$, an analog of coating thickness) from biomass burning is usually greater than 3 (Liu et al., 2017) and can be larger than 10 in remote sites (Wang et al., 2017a). Normally, when $R_{BC}$ is less than 1.5, it is probably from traffic sources, whereas secondary organic aerosol (SOA) dominated BC-containing particles is usually with a $R_{BC}$ greater than 4 (Lee et al., 2017). Moreover, the coating species can modify the hygroscopicity of BC-containing particles (Liu et al., 2013) when associated with hydrophilic materials, and some of them can serve as cloud condensation nuclei (CCN), therefore alter the albedo and precipitation of clouds indirectly (Dusek et al., 2010;Dusek et al., 2006).

In the past decades, a number of field studies on BC have been conducted in the winter of Beijing, and mainly focused on BC mass loadings, mixing states, optical properties, human health impacts and sources (coal combustion, biomass burning and vehicles, etc.) (Wu et al., 2017;Cheng et al., 2017;Ji et al., 2017;Wang et al., 2017b;Wu et al., 2016;Chen et al., 2016b;Meng et al., 2016;Wang et al., 2016b;Liu et al., 2016;Yang et al., 2014;Schleicher et al., 2013a;Schleicher et al., 2013b;Song et al., 2013;Zhang et al., 2017). There were real-time studies on BC, and on the chemical characteristics of total fine particles (including particles with and without BC) in Beijing. However, to the best of our knowledge, no study was conducted in real-time to characterize exclusively only BC-containing particles in Beijing despite the important effects of coating materials on BC properties aforementioned.





Currently, a few studies have explored BC-containing particles in other locations, e.g., Toronto (Willis
et al., 2016;Lee et al., 2015), California (Lee et al., 2017;Massoli et al., 2015;Cappa et al., 2012),
London (Liu et al., 2015) and Tibet (Wang et al., 2017a) by using the Aerodyne soot-particle Aerosol
Mass Spectrometer (SP-AMS) (Onasch et al., 2012;Lee et al., 2015;Wang et al., 2016a;Ge et al.,
2017b). The SP-AMS physically combines the 1064 nm laser vaporizer of single particle soot
photometer (SP2) into a high-resolution aerosol mass spectrometer (HR-AMS) (Onasch et al.,
2012;Canagaratna et al., 2007). After removal of the AMS tungsten vaporizer and by operating the
instrument with the laser vaporizer only, refractory BC as well as its associated coating can be
evaporated since the 1064 nm laser can selectively heat the BC (Massoli et al., 2015). In other words,
the laser-only SP-AMS can exclusively measure BC cores and the species coated on BC cores. This
unique technique allows us to explore in details the characteristics of BC-coating species with no
perturbations from other co-existing non-BC containing particles in ambient air.

Beijing, as the most reprehensive megacity with a large population in developing countries, the

BC-containing particles may have specific source profiles and physiochemical properties, therefore
elucidation of its characteristics is important to understand the haze formation and improve air quality
in such regions. In this work, we report for the first time the real-time measurement results on the
chemical composition, mass loading, size distribution, and sources/processes of the BC-containing
particles during wintertime of 2016 in urban Beijing by using the laser-only SP-AMS. Results
regarding mixing states and optical properties are presented in other publications in this special issue.

**2. Experiments**
**2.1 Sampling site and instrumentation**
As a part of the UK-CHINA Air Pollution & Human Health (APHH) winter campaign, we
conducted measurements at the Tower Division of Institute of Atmospheric Physics (IAP), Chinese
Academy of Science (39°58′N, 116°22′E) in Beijing (Figure S1 in the supplement), from 15 November
to 13 December of 2016. The site was surrounded by residential infrastructures and a freeway in the
east (360m).
The SP-AMS was deployed on the rooftop of Herong Building (~8m above the ground), with a
PM$_{2.5}$ cyclone (Model URG-2000-30EN) and a diffusion dryer in front of the inlet. The single particle
soot photometer (SP2, Droplet Measurement Technology, Inc., Boulder, CO, USA) was operated



simultaneously nearby inside another container (~20 m away) on the ground. The SP2 incandescence
signal was calibrated for BC mass by using Aquadag® black carbon standard (Aqueous Deflocculated
Acheson Graphite, Acheson Inc., USA) (Laborde et al., 2012). For the SP-AMS, since the filament
that ejects electrons can still heat the tungsten vaporizer up to ~200 °C (Willis et al., 2014) even it is
turned off, the tungsten vaporizer was thus physically removed to make sure only BC and its associates
were vaporized by the laser, and to eliminate influences from species uncoated on the BC cores.

The tuning and calibration procedures of the SP-AMS followed the procedures described

previously (Lee et al., 2015;Willis et al., 2016;Massoli et al., 2015;Wang et al., 2017a). During the
campaign, the SP-AMS was run with a 10-minutes cycle: one W mode with high chemical resolution
(2.5 min) and two mass sensitive V modes including one with particle time of flight (PToF) mode (2.5
min) and another one (5 min) with the measured $m/z$ up to 2000 to investigate fullerene-like carbon
clusters (Wang et al., 2016a). The filtered air measurement was performed for a day to determine the
detection limits (DLs) of various aerosol species and to adjust the fragmentation table. The ionization
efficiency (IE) and relative ionization efficiency (RIE) of sulfate and nitrate were calibrated by using
pure ammonium nitrate and ammonium sulfate according to Jayne et al. (2000), respectively. RIE of
BC was calibrated by using Regal Black (RB, REGAL 400R pigment black, Cabot Corp.) (Onasch et
al., 2012), and the average ratio of $C_1^+$ to $C_3^+$ was determined to be 0.53 to minimize the influence of
$C_1^+$ from non-refractory organics. However, it should be aware that laser-only SP-AMS cannot
vaporize ammonium nitrate/sulfate if they do not coat on BC, thus the IE and RIE calibrations were
done before removal of the tungsten vaporizer and the values were assumed to be unchanged after the
tungsten heater's removal (Willis et al., 2016). Note the RIE of BC was calibrated before the campaign
and was repeated in the middle and end of the campaign. RIEs of nitrate, ammonium, sulfate and BC
were determined to be 1.1, 3.82, 0.82, and 0.17, respectively. The default value of 1.4 was used as RIE
of organics (Canagaratna et al., 2007). Polystyrene latex (PSL) spheres (100-700 nm) (Duke Scientific
Corp., Palo Alto, CA) were used to calibrate the size before the campaign (Canagaratna et al., 2007) .
**2.3 Data Analysis**

Standard AMS data analysis software (Squirrel and Pika) based on Igor Pro 6.37 (Wavemetrixs,

Lake Oswego, OR, USA) were used to obtain the concentrations, mass spectra and size distributions
of BC and its coating species. All data were calculated based on high-resolution fitting results. Due to
different vaporization schemes between the SP-AMS and HR -AMS, mass spectra from these two





instruments even for the same population of particles are not entirely the same. Laser-only SP-AMS
can result in overall less fragmentation, therefore the mass profile may contain more large *m/z*
fragments and less small *m/z* fragments compared with that from HR-AMS (Massoli et al., 2015).
Therefore, here the elemental ratios of organics, i.e., oxygen-to-carbon, hydrogen-to-carbon and
nitrogen-to-carbon ratios (O/C, H/C and N/C) were determined by the Aiken approach first (Aiken et
al., 2008), and then O/C and H/C were corrected by using factors of 0.83 and 1.16, respectively
(Canagaratna et al., 2015).
Source apportionment for organics coated on BC was conducted by using Positive matrix
factorization (PMF) (Paatero and Tapper, 1994) Evaluation Tool written in Igor (Ulbrich et al., 2009).
In this study, high-resolution mass spectra (HR-MS) of organic (including BC) and inorganic species
were combined together to perform the PMF analyses (Sun et al., 2012;Wang et al., 2017a;Wang et al.,
2018). It should be noticed that, only fragment ions from polycyclic aromatic hydrocarbons (PAHs)
were included for *m/z* range of ~150 to ~250 in the PMF analysis because of the limited mass resolution
of SP-AMS. All PMF solutions were evaluated following the standard instruction (Zhang et al., 2011).
Finally, four types of organic aerosol (OA) associated with BC were determined eventually, including
a fossil fuel combustion OA (FFOA), a biomass burning OA (BBOA) and two oxygenated OA (OOA1
and OOA2).
Supporting data such as meteorological parameters including relative humidity (RH), wind speed
(WS), wind direction (WD) and temperature (T), as well as concentrations of gaseous species such as
$O_3$, $SO_2$, NO, $NO_2$, $NO_x$, $NO_y$, $NO_z$, and CO were measured in parallel simultaneously. All data
reported here were at local time (Beijing Time, UTC+8).

## 3. Results and discussions

### 3.1 Overview of BC-containing aerosol characteristics

Figs. 1 and 2 show the temporal variations of meteorology parameters, mass loadings of gaseous
pollutants (CO, $NO_x$, $SO_2$ and $O_3$), BC and its associated coating components (sulfate, nitrate,
ammonium, chloride, total OA and the four PMF-resolved OA factors). The campaign-averaged
composition of BC-containing particles and mass contributions of the four OA factors to total OA were
also displayed in Fig. 2. Overall, wind directions and speeds had close associations with the overall
mass loadings of BC-containing particles. The pollution periods (characterized by concentrations of





BC-containing particles above 10 µg m⁻³) were accompanied by low wind speeds (<4 m s⁻¹) and in a
relatively large part from southern air masses since Beijing is at the foot of the mountains which
facilitate the accumulation of pollutants from southern North China Plain (NCP). The clean periods
(characterized by the concentrations of BC-containing particles below 10 µg m⁻³) were mainly under
the control of strong winds (>4 m s⁻¹) from the northwest. During the campaign, the mass loadings of
*BC* cores and BC-containing particles ranged from 0.11 ~26.54 µg m⁻³ and 0.71~174.40 µg m⁻³, with
an average of 4.9 µg m⁻³ and 29.4 µg m⁻³, respectively. We also compared BC concentrations
determined by the SP-AMS with those from SP2, and they correlated quite well with each other ($r^2$ of
0.93; Fig. S3), indicating the quantification of BC by the SP-AMS is reliable.
The coating species occupied on average about 83.4% of the mass of BC-containing particles,
indicating BC was generally thickly coated throughout the whole campaign, with an average mass
ratio of coatings to BC ($R_{BC}$) of ~5. Organic aerosol (OA) was the most abundant coating component,
taking up to 59.4% of the total mass, followed by nitrate, sulfate, ammonium and chloride (8.8%, 6.5%,
4.7% and 4.0%, respectively). OA correlated quite well with BC ($r^2$ of 0.97), suggesting that many OA
species were co-emitted and mixed with BC, and indeed, primary OA (POA=FFOA+BBOA) was
found to dominate the OA mass (66.3%=43.9%+22.4%). Chloride (Cl⁻) had a great correlation with
*BC* ($r^2$ of 0.94), suggesting it was mainly associated with primary emissions, for example, gasoline,
diesel and coal combustion during wintertime in urban Beijing. Sulfate and nitrate are typically
secondarily formed, therefore their correlations with BC were relatively weak ($r^2$ of 0.64 for $SO_4^{2-}$ *vs.*
BC, and 0.60 for $NO_3^-$ *vs.* BC). Their properties are discussed in more details in the following sections.

**3.2 Chemically-resolved size distributions of BC-containing particles**
Fig. 3a shows the campaign-averaged mass-based size distributions of major BC-coating species,
including organics (BC-Org), sulfate (BC-Sulfate), nitrate (BC-Nitrate), chloride (BC-Chl) and *BC*
core itself. It should be noticed that the size distribution of BC was scaled from that of $m/z$ 24 ($C_2^+$),
as other major carbon cluster ions might be significantly affected by other ions, for example, $C_1^+$ at
$m/z$ 12 can be influenced by fragments from non-BC organics, $C_3^+$ at $m/z$ 36 by $HCl^+$, $C_4^+$ at $m/z$ 48 by
$SO^+$, and $C_5^+$ at $m/z$ 60 by $C_2H_4O_2^+$ etc. Similarly, the size distribution of BC-Chl was scaled from $Cl^+$
signal at $m/z$ 35. As shown in Fig. 3a, on average, size distributions of BC-Sulfate, BC-Nitrate and BC-
Org displayed a similar pattern with a major peak at ~550 nm (vacuum aerodynamic diameter, $D_{va}$),



suggesting that they were relatively well internally mixed. However, the BC presented a remarkably
different pattern with a much broader distribution and smaller peak sizes than its coating species, and
in particular, relatively small particles tended to have thin coatings.
Figs. 3b-f further present image plots of size distributions of the major aerosol components as a
function of $R_{BC}$ (as a surrogate for coating thickness). Different from the average data shown in Fig.
3a, the coating species can be roughly classified into two modes separated by $R_{BC}$ of ~4.5. Most sulfate
and nitrate concentrated at $R_{BC}$>4.5 (Figs. 3b and 3c): Sulfate peaked in a narrow $R_{BC}$ range of 5.5~6.5,
while significant nitrate mass could distribute across a wider $R_{BC}$ range (even to $R_{BC}$ of ~8). Only
organics and chloride had a significant portion of mass distributed on relatively thinly coated *BC*-
containing particles at $R_{BC}$<4.5 (Figs. 3e and 3f). Specifically, they both showed a sub-mode locating
in the regime with $R_{BC}$ of ~3.5-4.5 and $D_{va}$ of ~200-700nm. These sub-modes suggest that organics or
chloride are partially from primary sources as freshly emitted BC are more likely thinly coated. This
is consistent with that organics included species from fossil fuel and biomass burning combustion
revealed by the PMF analysis. Similarly, coal burning might contribute to chloride during wintertime
in Beijing (Sun et al., 2016). As for sulfate and nitrate, since they are predominantly secondary species,
they would coat on BC cores due to chemical aging therefore mostly distributed at higher $R_{BC}$.

**3.3 Sources of organic coating species**
The high-resolution mass spectra (HRMS) of different factors of the organic coating resolved from
PMF analyses, their relative contributions and diurnal cycles of temporal variations relative to BC are
shown in Fig. 4. Fig. 4a illustrates the mass profile of the fossil fuel combustion OA with BC carbon
clusters (FFOA + BC). This factor had a low O/C ratio of 0.16. In this work, this factor might include
emissions from both traffic and coal combustion, as it contained a series of significant PAHs ion
fragments in the mass spectrum (PAHs fragments are negligible in other factors) indicative of coal
burning (Sun et al., 2014;Sun et al., 2016), and presented a good correlation with $C_4H_9^+$ ($r^2$ of 0.72)
which is a AMS tracer ion of vehicle emissions (Zhang et al., 2005). Temporal variations of FFOA also
correlated well with $C_9H_7^+$ (*m/z* 115, $r^2$ of 0.92) and $Cl^-$ ($r^2$ of 0.60), which have been proposed as
possible coal combustion tracer species (Yan et al., 2018;Sun et al., 2014). The FFOA/BC (Fig. 4f)
appeared to be higher during nighttime than that during daytime. Note the diurnal pattern of BC itself
(Fig. 5c) was similar as that of FFOA/BC. The diurnal variations of BC might be influenced by both



fossil fuel combustion activities and relatively low PBLH during nighttime. The fossil fuel combustion
included coal burning and vehicle emissions (gasoline cars, and the heavy-duty diesel vehicles that are
only allowed to enter the city during later night). The mass ratios of different factors to BC shall have
less influences from PBLH, therefore high levels of FFOA/BC strongly indicate that co-emitted
organic species with BC from fossil fuel combustion were enhanced during nighttime.

Figure 4b shows the mass spectrum of BBOA and related BC clusters. One feature of this factor

is that it had relatively high fractional contributions of $C_2H_4O_2^+$ (1.47% of total) and $C_3H_5O_2^+$ (0.95%),
which are often regarded as AMS marker ions from biomass burning emitted levoglucosan (Cubison
et al., 2011;Mohr et al., 2009). Note the FFOA also contained appreciable $C_2H_4O_2^+$ and $C_3H_5O_2^+$
signals, partially due to that coal burning (such as lignite) can emit some levoglucosan as well (Yan et
al., 2018). Nevertheless, mass fractions of $C_2H_4O_2^+$ and $C_3H_5O_2^+$ in FFOA were less than those in
BBOA, and they correlated much better with BBOA than those with FFOA (for examples, $r^2$ of 0.90
for BBOA *vs.* $C_2H_4O_2^+$, and 0.72 for FFOA *vs.* $C_2H_4O_2^+$). The BBOA correlated very well with another
biomass burning tracer - $K^+$ ($r^2$ of 0.90). In addition, BBOA had negligible PAHs ion fragments while
the FFOA contained remarkably high PAHs signals. Such characteristics are generally in agreement
with previous AMS findings in the same location during wintertime in Beijing (Sun et al., 2016). For
these reasons, the second factor was identified as BBOA. The diurnal pattern of BBOA/BC reached
minimum during afternoon and was overall high during nighttime, similar as FFOA/BC, indicating the
nighttime enhancement of BB-related organics emissions in wintertime Beijing.

Besides the two POA factors, we also identified two secondary OA factors (OOA1 and OOA2),

whose O/C ratios were 0.45 and 0.28, respectively. OOA1 was the most oxidized OA factor that had a
higher $CO_2^+/C_2H_3O^+$ ratio than that of OOA2. The correlation between OOA1 and sulfate was better
than it with nitrate ($r^2$ of 0.99 *vs.* 0.86). As a comparison, the less oxygenated OOA2 correlated better
with nitrate than it with sulfate ($r^2$ of 0.59 *vs.* 0.34). These characteristics are consistent with previous
AMS-PMF results (Zhang et al., 2011). Opposite to the diurnal cycles of FFOA and BBOA, the
OOA2/BC ratio arose significantly from early morning and peaked in the afternoon (~3pm). The
diurnal pattern of OOA1/BC presented a similar peak at ~3pm. This result demonstrates a clear
evidence and important role of photochemical reactions to the formation of secondary organic species.
However, the precursors leading to the formations of OOA1 and OOA2 remain to be elucidated.
Interestingly, for OOA1/BC, in addition to the peak during afternoon, the ratio increased during early





evening and remained at high levels until early morning. This result indicates that nighttime aqueous-
phase processing (high levels of RH during nighttime shown in Fig. 5a) can also contribute to OOA1
production. As such behavior was not observed for OOA2/BC, it agrees with previous field and
laboratory findings that aqueous-phase reactions tend to produce more highly oxygenated species
(Ervens et al., 2011;Ge et al., 2012;Herrmann et al., 2015;Xu et al., 2017).

Overall, the mass fractions of BC cores that were associated with fossil fuel combustion, biomass

burning, less and more oxygenated secondary processes were 32.7%, 31.8%, 18,7% and 16.9%,
respectively (Fig. 4e). The BC was predominantly coated by primary species.

**3.4. Diurnal patterns of BC and coating species**

Fig. 5 presents the diurnal cycles of meteorological parameters (T, RH, WS and WD), BC

concentrations and $R_{BC}$, mass ratios of major species to BC, gaseous species (CO, $SO_2$ and $NO_x$), O/C
and $OS_c$ (oxidation state, defined as 2*O/C-H/C)(Kroll et al., 2011). Note BC did not present a peak
at 8:00 am, yet $R_{BC,}$ Org/BC, $SO_4^{2-}$/BC, $NO_3^-$/BC and $Cl^-$/BC were all low at ~8:00 am. This was likely
attributed to increase of the mass fractions of fresh and barely coated $B$C-containing particles (rather
than the increase of absolute concentrations of fresh BC-containing particles) emitted during morning
rush hours from traffic emissions, etc. This was consistent with the decreases of O/C and $OS_c$ and
increases of CO and $NO_2$ at 8:00 am of the day. On the contrary, the $R_{BC}$ drop at ~4:00 pm was unlikely
due to influences of afternoon rush hours, as there were no increases of CO, $NO_2$, and both O/C and
$OS_c$ were at high levels. In fact, the 4:00pm $R_{BC}$ drop was mainly caused by the large decrease of
organics coating concentration - mainly fossil fuel and biomass burning OA (Fig. 4f).

The diurnal variation of $NO_3^-$/BC peaked at ~3-4 pm, consistent with the variation of T, and similar

as those in the previous reports during wintertime in Beijing (Ge et al., 2017a;Sun et al., 2016),
reflecting the dominated contribution of photochemical formation of nitrate. $SO_4^{2-}$/BC showed a
relatively small afternoon increase, indicating partial sulfate was produced from photochemical
activities; it also presented a nighttime enhancement, similar as OOA1/BC, suggesting the sulfate
formation in aqueous-phase, consistent with the nighttime increase of RH and decrease of temperature
(Fig. 5a). Due to increases of FFOA/BC, BBOA/BC and OOA1/BC (the portion likely from aqueous-
phase production), Org/BC remained at high levels during nighttime. All these increases added together,
leading to the high $R_{BC}$ during nighttime. In addition, $Cl^-$/BC varied generally similar to those of





FFOA/BC and BBOA/BC, again indicating its strong association with primary emissions.

### 3.5 Characteristics of coating species during different periods

### 3.5.1 Coating compositions at clean and pollution periods

Fig. 6 shows the variations of BC-coating compositions as a function of coating thickness during
clean (CP) and pollution periods (PP) (separated by the concentration of 10 µg/m$^3$), respectively.
Contrasting difference of the coating composition during these two cases was observed: primary OA
(especially FFOA) appeared to be the most abundant component during CP while mass contributions
of secondary organic and inorganic species were remarkably high during PP (Figs. 6a and b), and the
average $R_{BC}$ during PP (~5.1) was also higher than that during CP (~4.5) (Fig. 6f). These results again
reinforce the importance of secondarily formed species to the heavy haze pollution in urban Beijing
(Huang et al., 2014). Furthermore, the BC coating composition as well as OS$_c$ during CP were both
relatively stable against $R_{BC}$ (Fig. 6c). On the contrary, during PP, with the increase of $R_{BC}$, the mass
fractions of secondary species (OOA1, nitrate and sulfate) increased clearly, especially at $R_{BC}>5$;
consistently, OS$_c$ of organic coating increased from ~-0.85 to >-0.70. Such behavior again highlights
the contribution of chemical aging process to the heavy haze pollution.
Relative to other observations (Wang et al., 2017a;Massoli et al., 2015;Cappa et al., 2012), the
levels of $R_{BC}$ during both CP and PP are much smaller than those for highly aged BC, which might
have $R_{BC}>10$. This can be expected for urban aerosols. On the other hand, the $R_{BC}$ levels here are also
generally higher than those found for the BC-containing particles in Los Angeles where the average
$R_{BC}$ was typically smaller than 4 due to predominant influence of vehicle emissions (Lee et al., 2017).
Regarding the variations of coating composition *vs.* $R_{BC}$, the behavior during PP is in fact consistent
with a few previous field measurement results in American or European urban locations (Massoli et
al., 2015;Liu et al., 2017;Lee et al., 2017;Cappa et al., 2012;Collier et al., 2018), indicating a general
behavior for BC-containing particles in urban area that more aged BC tends to have thicker coating.
Yet this property can be altered if significant POA emissions exist, such as the case during CP in this
work, and a case with heavy BBOA influences observed in Tibet Plateau (Wang et al., 2017a).

### 3.5.2 Coating compositions at two different episodes

Although we demonstrated in Section 3.5.1, the heavy pollution of BC-containing particles was



on average associated with more secondary species, the underlying governing factors of individual
pollution events might vary from each other. Here we investigated the characteristics of BC-containing
particles in two most polluted episodes occurring during the campaign. The first episode (FE) was
accompanied with relatively high RH (from 6:00 pm of 3 Dec. to 8:00 am of 4 Dec., 2016), while the
second episode (SE) was dominated by primary emissions (from 0:00 am to 6:00 am of 11 Dec., 2016).
The average mass loadings of BC cores and BC-containing particles were 18.1 μg m$^{-3}$ and 123.1 μg
m$^{-3}$ during FE, 14.4 μg m$^{-3}$ and 80.0 μg m$^{-3}$ during SE, respectively - both were much higher than the
campaign-averaged BC of 4.9 μg m$^{-3}$ and BC-containing particles of 29.4 μg m$^{-3}$.

For FE, the average T and RH were ~4.2 °C and ~78%, respectively. The average T was close to

the campaign-average value of 4.8ºC, but the air was more humid than the campaign-average RH of
~50%. Correspondingly, we observed clear increases of the mass contributions of sulfate from 6.5%
to 10.3%, nitrate from 8.8% to 10.2%, OOA1 from 7.5% to 11.5% (Figs. 7a and 7c). Such
enhancements were very likely linked with the aqueous-phase processing as this episode occurred
during nighttime and was characterized with high RH conditions. During FE, nitrate and sulfate also
correlated very well ($r^2$ of 0.94; Fig. S4), therefore formation of nitrate would also relate with aqueous-
phase processing in this episode. As a comparison, the mass fraction of photochemical-relevant OOA2
decreased significantly from campaign-average 13.3% to 9.8%. In addition, mass fraction of Cl$^-$ also
increased from campaign-average 4.0% to 5.3%; meanwhile, we found that relative to the campaign-
average values, the KCl$^+$/BC ratio decreased 14%, the K$_3$SO$_4^+$/BC ratio increased 28%, possibly
indicating that the heterogeneous replacement reactions of coal-burning related Cl$^-$ by SO$_4^{2-}$ during FE
(Fig. S4). Overall, due to mainly the aqueous-phase production of secondary coating components,
comparing to campaign-average values, the average $R_{BC}$ during FE became larger (5.5 $vs.$ 5.0), the OA
became more oxygenated (O/C of 0.18 $vs.$ 0.15), and size distributions of OA, sulfate and nitrate all
shifted to larger peak sizes (Fig. S5a).

On the other hand, for SE, even though it also occurred during nighttime, the average RH was

significantly low (~47%), and it was overwhelmingly dominated by primary species (50.6% of FFOA,
15.2% of BBOA and 18% of BC). Secondary sulfate and nitrate only occupied 2.5% and 2.2% of the
total mass of BC-containing particles. Nighttime aqueous-phase related OOA1 contribution was nearly
negligible (only 0.8%), which in another way, manifests that nighttime efficient OOA1 production was
strongly associated with high RH conditions. Due to the contribution of fresh primary emissions, the



coating OA was less oxygenated than that of campaign-averaged OA (O/C of 0.12 *vs.* 0.15), and the average $R_{BC}$ during PE was consistently smaller (4.5 *vs.* 5.0). Mass spectrum of BC-Org (Fig. 7b) also contained significant PAHs fragments, in line with the large contribution from FFOA (mainly coal combustion). Average size distribution of OA during SE was broader and peaked in a smaller diameter (<500 nm $D_{va}$) (Fig. S5b), in response to the dominance of POA. Occurrence of the highly polluted SE demonstrates that even though the pollutions of BC-containing particles in urban Beijing during winter are on average governed by secondary species, local primary emissions sometimes can lead to serious and short-term pollution events as well.

## 4. Conclusions

As part of the UK-China 2016 APHH winter campaign, for the first time, an Aerodyne SP-AMS was introduced to exclusively determine the chemical compositions of BC-containing particles in urban Beijing. We found the average concentrations of *BC* and its coating species were 4.9 and 24.5 μg m$^{-3}$, namely the $R_{BC}$ (mass ratio of coating to *BC*) was ~5.0. The coating was dominated by organics (59.4% of total mass of BC-containing particles), followed by nitrate and sulfate (15.3% together). Size distribution data demonstrate that larger BC-containing particles tend to have thicker coating, more secondary species and more internally mixed coating components. PMF analyses further identified two POA factors related with fossil fuel and biomass burning, respectively, which dominated the total OA mass. Two SOA factors were also separated, and both of them were found to be mainly contributed by photochemical activities, besides a fraction of the highly oxidized OA factor could be produced by nighttime aqueous-phase reactions. In addition, significant photochemical formation of nitrate rather than sulfate was observed.

Comparisons of the coating compositions between clean and pollution periods shows the critically important role of chemical aging for the pollution of BC-containing particles in urban Beijing. We also found that in one case, aqueous-phase production might lead to serious pollution under high RH conditions, while in another case, fossil fuel combustion could cause extreme and short-term heavy pollution. Comparisons between the BC-containing particles and the total submicron aerosol particles during this campaign will be presented in details in near future.

## Acknowledgements



This work was supported by the National Key R&D program of China (2016YFC0203501), the
Natural Science Foundation of China (21777073, 91544220, 21577065, and 41571130034,), and the
International ST Cooperation Program of China (2014DFA90780), the UK Natural Environment
Research Council (grant ref. NE/N00695X/1).

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



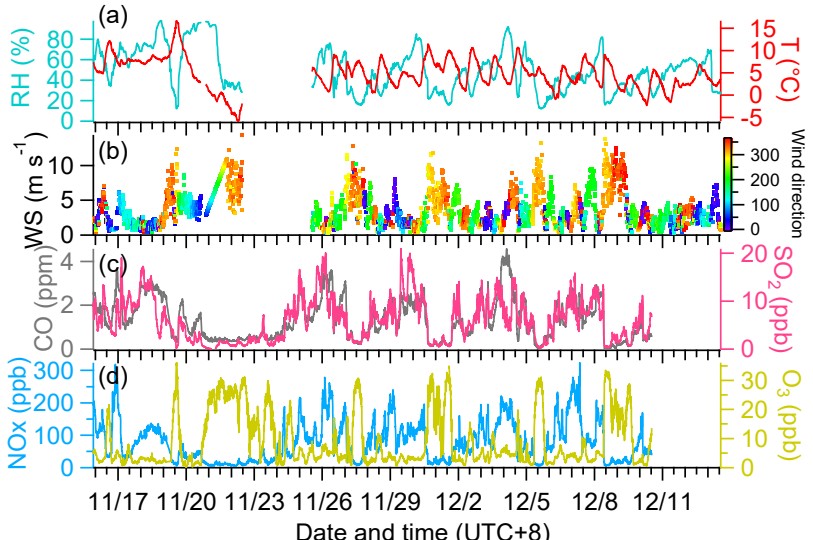


Figure 1. Temporal variation of (a) relative humidity (RH) and temperature (T, °C), (b) wind speed (WS, m s⁻¹) and wind
direction (WD), and (c)(d) mass loadings of CO, SO₂, NOₓ and O₃.







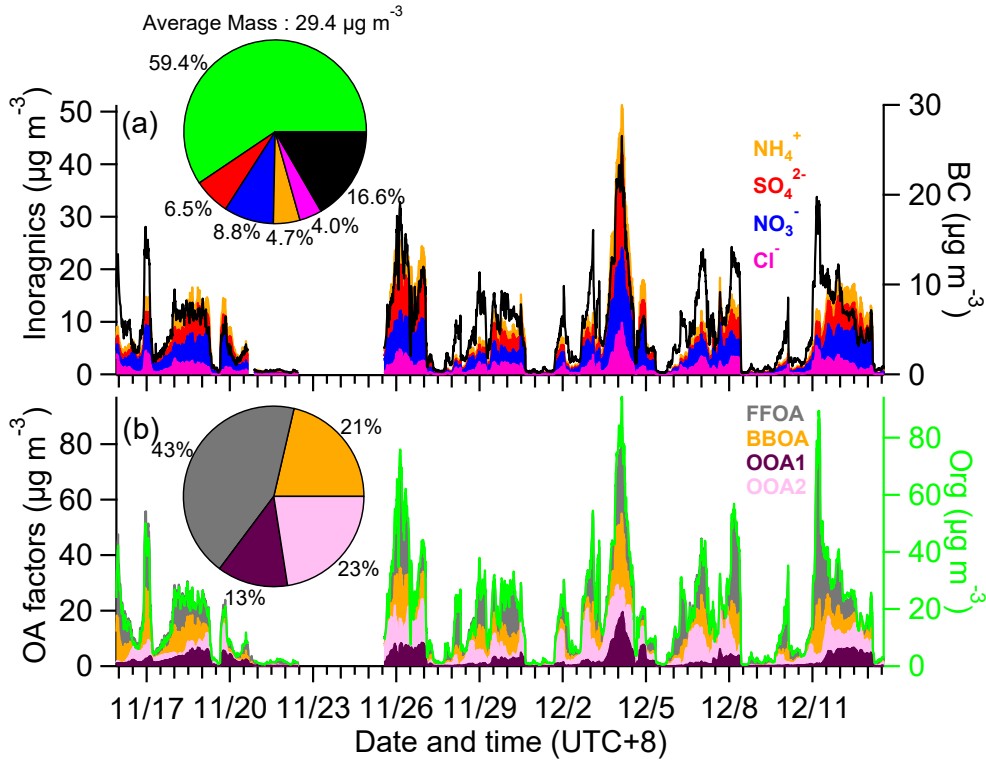


Figure 2. (a) Temporal variations of mass loadings of inorganic coating components (sulfate, nitrate, ammonium and
chloride) and BC cores, and (b) temporal variations of mass loadings of organic coating (Org) and PMF separated OA
factors (inset pie charts show the average composition of total BC-containing particles, and organics, respectively).



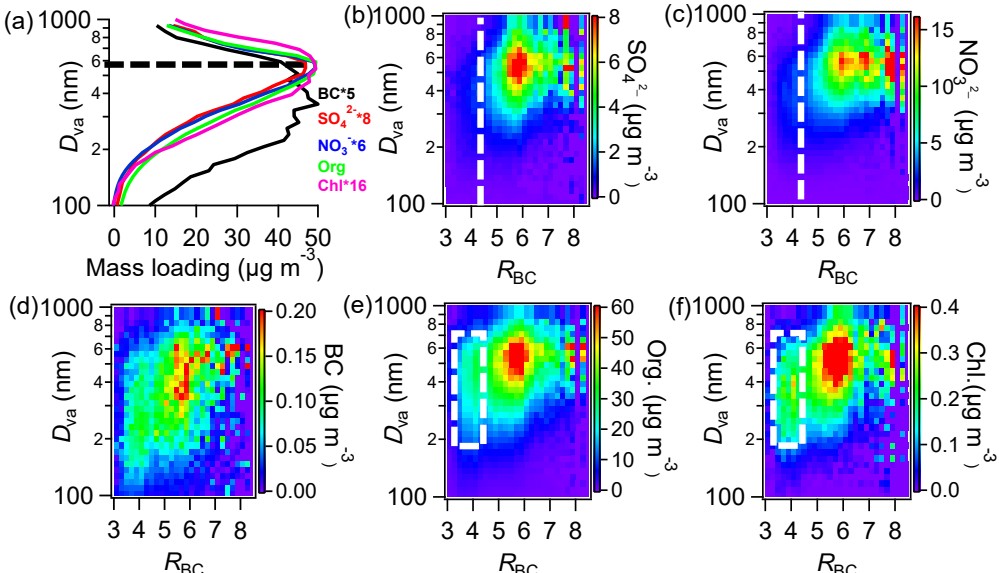


Figure 3. Mass-based campaign-averaged size distributions: (a) major coating components and BC cores, and (b-f) image
plots of size distributions of sulfate, nitrate, BC, organics, and chloride as a function of $R_{BC}$ (mass ratio of coating-to-$BC$)
(Note size distributions of BC and chloride were scaled from those of $m/z$ 24 and $m/z$ 35, respectively)




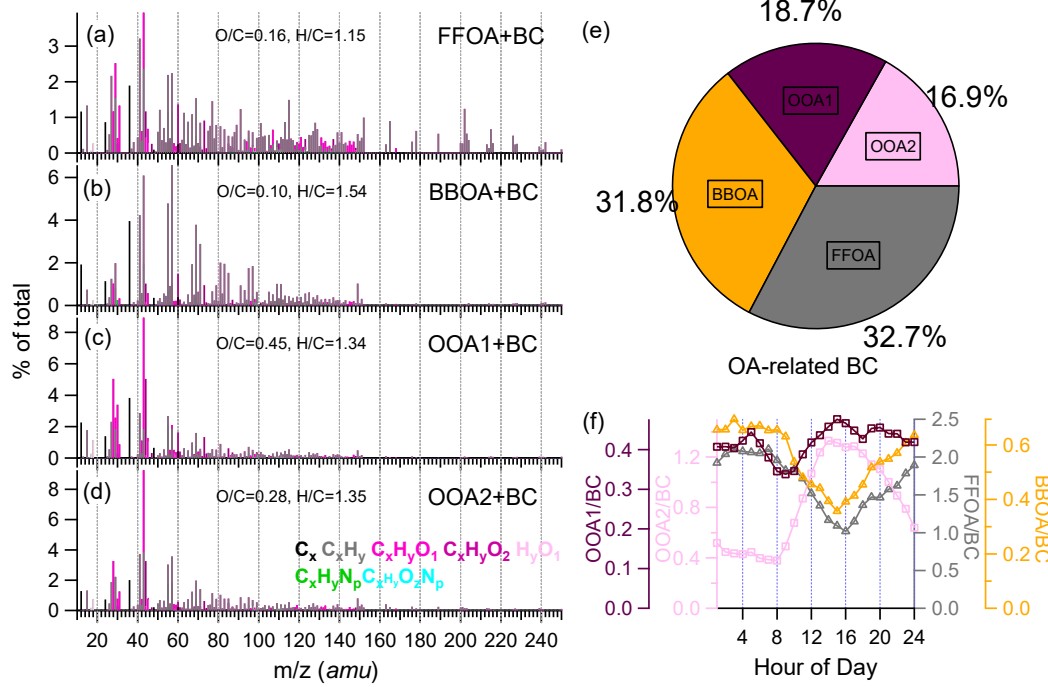


Figure 4. High-resolution mass spectra of (a) fossil fuel combustion OA (FFOA + BC), (b) biomass burning OA (BBOA + BC), (c) OOA1 + BC, (d) OOA2 + BC, (e) Mass fractions of the BC fragments apportioned in different OA factors, and (f) diurnal cycles of the four OA factors relative to BC.






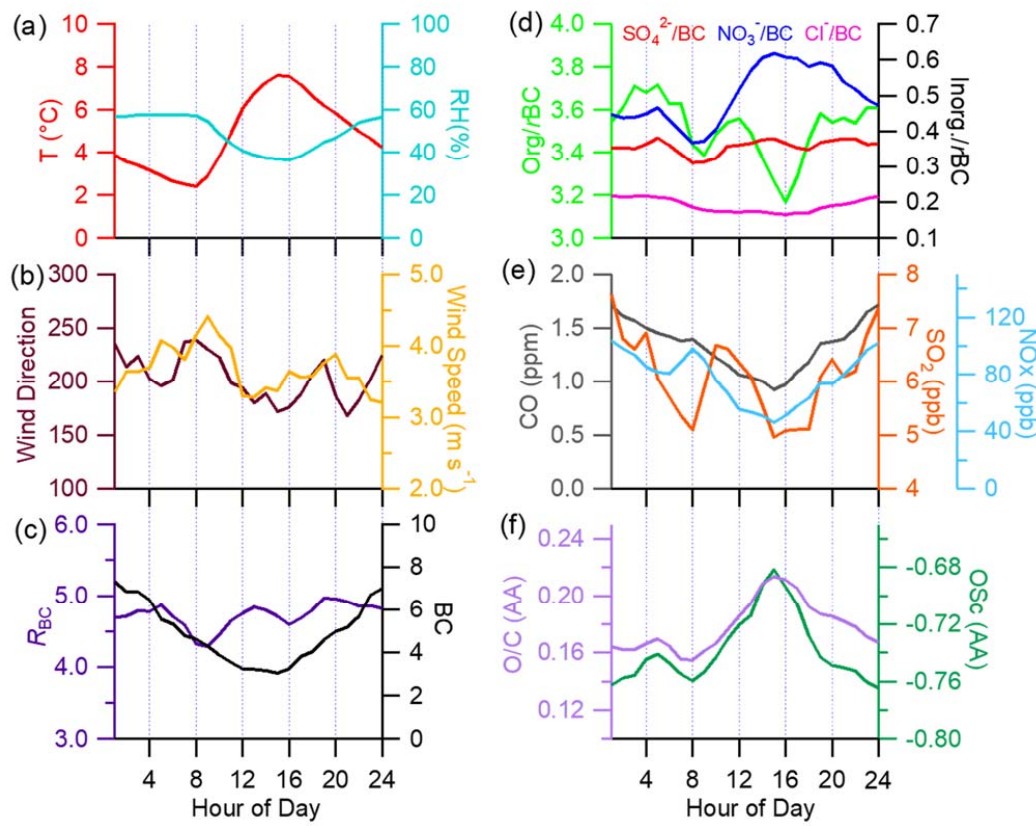


Figure 5. Diurnal cycles of (a) T and RH, (b) wind direction and wind speed, (c) mass ratio of coatings to BC ($R_{BC}$) and
BC, (d) Org/BC, $SO_4^{2-}$/BC, $NO_3^-$/BC and $Cl^-$/$BC$, (e) mass loading of gaseous species (CO, $SO_2$, $NO_x$), and (f) O/C and
oxidation state ($OS_c$=2*O/C-H/C).




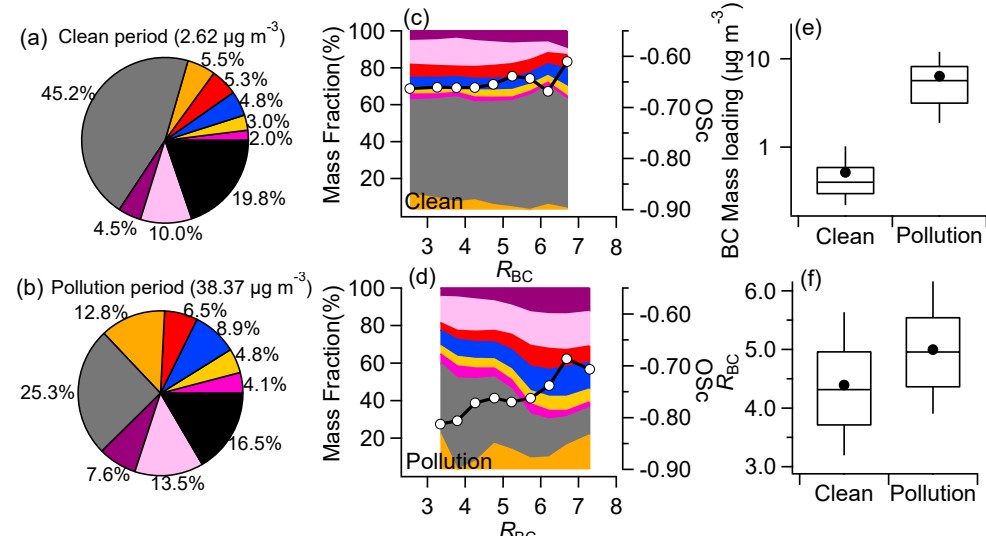


Figure 6. (a)(b) Average compositions of BC-containing particles during clean and pollution periods, (c)(d) mass fractions
of the non-BC coating components (left $y$-axis) and $OS_c$ (right $y$-axis) during clean and pollution periods as a function of
$R_{BC}$, box plots of BC mass loadings (e) and $R_{BC}$ during clean and pollution periods (colors of the components are consistent
with those in Fig. 2).




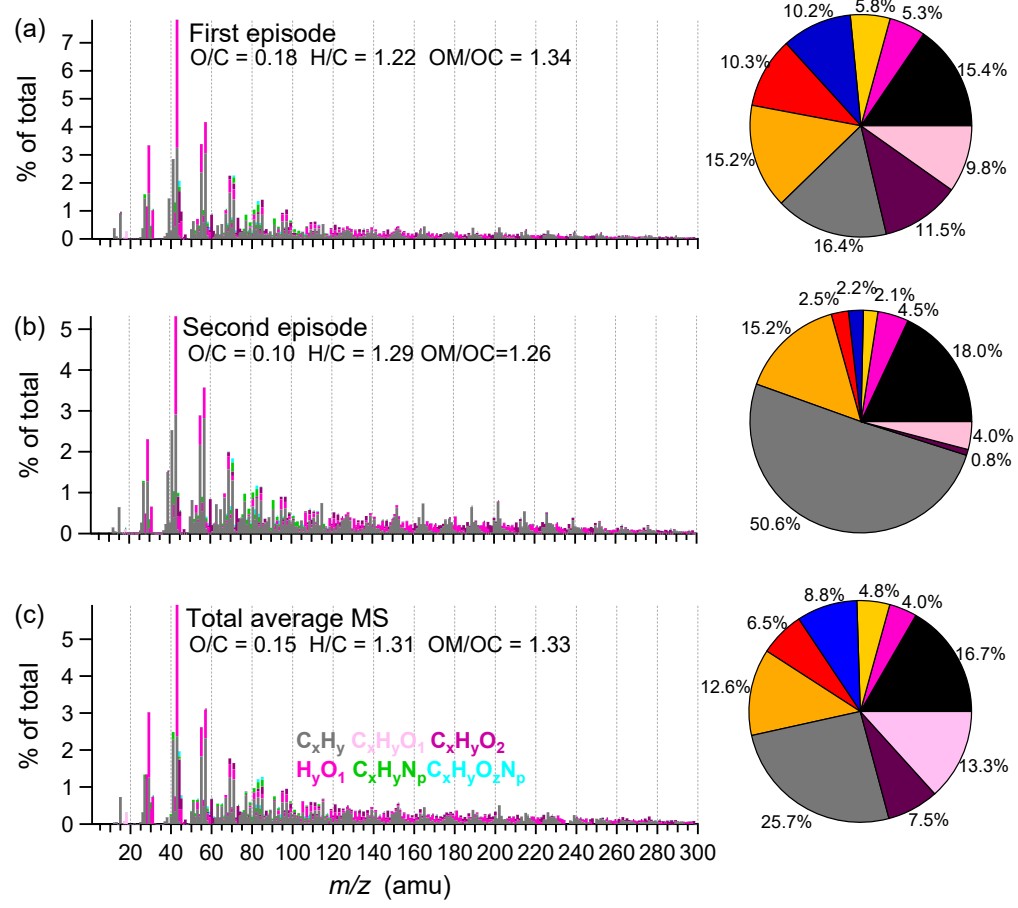


Figure 7. High-resolution mass spectra of the average OA at different episodes: (a) first episode (FE), (b) second episode
(SE) and (c) whole campaign (inset pies show the average compositions during corresponding episodes; colors of
different components are consistent with those in Fig. 2).

