# Peer review of "Characterization of black carbon-containing fine particles"

_Atmospheric Chemistry and Physics, 2018_

## Referee Comment (RC1) · Anonymous Referee #2 · 19 Sep 2018

This manuscript belongs to the results of APHH 2016 winter campaign, it reports measurement results on the chemical properties of black carbon and the coating materials on the black carbon cores. It used a specific Aerodyne soot-particle aerosol mass spectrometry, which allows to analyze exclusively black-carbon containing particles. This technique avoids interferences from other non-BC containing particles, therefore can elucidate more accurately and directly the properties and evolution of BC in ambient air. Such type of measurement was for the first time conducted in wintertime Beijing, the data and results are thus very valuable. I agree its publication in ACP after some minor revisions suggested below: (1) Is there any review paper to introduce this APHH campaign, and how does this paper contribute to the overall goal of this campaign? It should be mentioned. (2) The details of PMF analysis results were not

mentioned, a diagnostic plot can be provided, at least in the attachment, to justify the choice of PMF solution. (3) Did the authors observe fullerene-related carbon cluster ions in the mass spectra of Beijing BC aerosols? (4) Some modifications on the figures are necessary. For example, Font sizes in Fig 4 appear to be small; mass spectra in Fig.7 are less clear. (5) In Fig.6, the RBC ranges for clean and pollution periods are different, it is better to compare the variations on the same scale? (6) One general suggestion is that this dataset is unique as it measures only BC-particles in a highly polluted environment, the reviewer feels the discussion needs more comparisons with results in other locations or environments. As mentioned by the authors, such measurements were conducted in other sites in US, and Europe, even it is very scarce in China. This can help to show what is special or different or important of the findings observed in Beijing, and what are the implications of such findings to the atmospheric chemistry.

---

## Referee Comment (RC2) · Anonymous Referee #1 · 1 Nov 2018

Characterization of black carbon-containing fine particles in Beijing during wintertime

Wang et al.,

BC also called as soot is an important aerosol from incomplete combustion of fossil fuels and biomass burning. Understanding the soot mixing state in polluted air of Beijing, it is quite important issue to evaluate their potential optical, hygroscopic, and human health. The authors used one SP-AMS to determine mixing state of soot particles collected in Beijing during the wintertime. They found coating/BC ratio was at 5.0, much smaller than highly aged soot in other places. Also, they studied coating chemical species and their possible formation mechanism. The scope of this study is suitable for ACP. However, the paper need to one substantial revision before it can be published. I list several concerns about the conclusions

(1) L28-29 deleted very (2) L30 only (3) L33-34 how do the result indicate dominant contributions from primary emissions? You can say that these particles might source from local emissions instead of long-range transport particles. Am I right? (4) L35-36, 38-40, seemly for me, the conclusion is constrast. One you mentioned primary emission. Other one you want to mention the secondary species. (5) L41 at-during (6) L44-45, I don't think the conclusion is from your solid result. Most you speculate these results. (7) L44-47 the conclusion cover all the possible. I would ask the author revise it carefully. What is your conclusions during the sampling period. If these solid conclusion are not from this study, you need to remove it. Seemly, I like to see what you find on BC particles not for haze formation. (8) L56 Morphology of BC might be altered greatly. These citations don't supply any morphology of BC particles. You need find others from electron microscopies. (9) L77, I don't agree with the claim. For example, Wu et al., 2017. Size distribution and source of black carbon aerosol in urban Beijing during winter haze episodes. Atmos. Chem. Phys. 17 (12), 7965-7975. The study seemly, give the online BC-containing particles in Beijing. (10) L161 Discussion, deleted s (11) L260-261, L284-285, L331-332 L347-348, all the parts discussed the aqueous reactions for nitrate and SOA formation during the nighttime. I take a look at the data from the study. It is too simply to get such conclusion. I might ask the authors cite more related references here. For example, Wu et al., 2018. Environmental Science & Technology Letters 5 (3), 160-166;Sun et al., 2018. Journal of Geophysical Research: Atmospheres 123 (2), 1234-1243.Kuang et al., 2016. Geophysical Research Letters 43 (16), 8744-8750. (12) L278-279 I don't understand the sentence. Why was the large decrease of organics coating concentration? (13) L292 at-during (14) L307 This can be expected for urban aerosols. I don't understand it. Why? (15) L328 of-at (16) L269 miss comma after ws (17) L317 at two polluted episoes (18) For section 3.5.2 Seemly, the authors found different coating species on soot particles. FE, the author found large SOA; SE the author proposed large POA instead of SOA. Do the authors answer how POA associated with BC? If these particles were emitted from sources, these mixing should occur in all the time, not just SE. Were there different sources in

SE and FE? Seemly, the author didn't supply any wind and backtrajectories here. I would ask the authors carefully check the data. Make sure the differences in FE and SE are large. Here the authors only compared the organics. What about the sulfate and nitrate are in the coating of BC there. I am certainly struggling on the part.

---

## Author Comment (AC1) · 19 Dec 2018

**Response to Reviewer's Comments**

**Manuscript Number**:  acp-2018-800

**Authors:** Junfeng Wang, Dantong Liu, Xinlei Ge, Yangzhou Wu, Fuzhen Shen, Mindong Chen, Jian Zhao, Conghui Xie, Qingqing Wang, Weiqi Xu, Jie Zhang, Jianlin Hu, James Allan, Rutambhara Joshi, Pingqing Fu, Hugh Coe and Yele Sun

**Response to Reviewer #1**

This manuscript belongs to the results of APHH 2016 winter campaign, it reports measurement results on the chemical properties of black carbon and the coating materials on the black carbon cores. It used a specific Aerodyne soot-particle aerosol mass spectrometry, which allows to analyze exclusively black-carbon containing particles. This technique avoids interferences from other non-BC containing particles, therefore can elucidate more accurately and directly the properties and evolution of BC in ambient air. Such type of measurement was for the first time conducted in wintertime Beijing, the data and results are thus very valuable. I agree its publication in ACP after some minor revisions suggested below:

**Authors' reply:** We thank the reviewer for his very positive feedback, and our point-to-point replies to the reviewer's comments are listed below.

(1) Is there any review paper to introduce this APHH campaign, and how does this paper contribute to the overall goal of this campaign? It should be mentioned.

**Authors' reply:** Yes, by the time of submission, the overview paper is not yet posted. It is now mentioned in the manuscript: Shi, et al.: Introduction to Special Issue – In-depth study of air pollution sources and processes within Beijing and its surrounding region (APHH-Beijing), Atmos. Chem. Phys. Discuss., https://doi.org/10.5194/acp-2018-922, 2018.

(2) The details of PMF analysis results were not mentioned, a diagnostic plot can be provided, at least in the attachment, to justify the choice of PMF solution.

**Authors' reply:** As requested by the reviewer, we have added a diagnostic plot of the PMF results as Fig. S2 in the supplement to justify the PMF results.

(3) Did the authors observe fullerene-related carbon cluster ions in the mass spectra of Beijing BC aerosols?

**Authors' reply:** Yes, we have observed fullerene-related carbon clusters during this campaign. In fact, we will publish the observation results regarding fullerenes from Beijing and also other sites in another paper, therefore we did not include them here

(4) Some modifications on the figures are necessary. For example, Font sizes in Fig 4 appear to be small; mass spectra in Fig.7 are less clear.

**Authors' reply:** As suggested, we have modified the figures (especially Figs. 4 and 7) to have a better quality.

(5) In Fig.6, the RBC ranges for clean and pollution periods are different, it is better to compare the variations on the same scale?

**Authors' reply:** In Figs.6c and 6d, the $R_{BC}$ ranges are set on the same scale.

(6) One general suggestion is that this dataset is unique as it measures only BC-particles in a highly polluted environment, the reviewer feels the discussion needs more comparisons with results in other locations or environments. As mentioned by the authors, such measurements were conducted in other sites in US, and Europe, even it is very scarce in China. This can help to show what is special or different or important of the findings observed in Beijing, and what are the implications of such findings to the atmospheric chemistry.

**Authors' reply:** Thanks for the suggestion. We in fact have included some comparisons with previous results including in Los Angeles, London, etc. during our discussion. Per the request, we have made some necessary minor changes in revising the manuscript. Please see the modified version for details.

**Response to Reviewer #2**

BC also called as soot is an important aerosol from incomplete combustion of fossil fuels and biomass burning. Understanding the soot mixing state in polluted air of Beijing, it is quite important issue to evaluate their potential optical, hygroscopic, and human health. The authors used one SP-AMS to determine mixing state of soot particles collected in Beijing during the wintertime. They found coating/BC ratio was at 5.0, much smaller than highly aged soot in other places. Also, they studied coating chemical species and their possible formation mechanism. The scope of this study is suitable for ACP. However, the paper need to one substantial revision before it can be published. I list several concerns about the conclusions.

**Authors' reply:** We thank the reviewer for his valuable comments, and our point-to-point replies to the reviewer's comments are listed below.

(1) L28-29 deleted very (2) L30 only

**Authors' reply: Done**

(3) L33-34 how do the result indicate dominant contributions from primary emissions? You can say that these particles might source from local emissions instead of long-range transport particles. Am I right?

**Authors' reply:** We agree that this may not be appropriate, and we deleted this sentence.

(4) L35- 36, 38-40, seemly for me, the conclusion is contrast. One you mentioned primary emission. Other one you want to mention the secondary species.

**Authors' reply:** We agree with that the description is not very specific. In line with comment (3), we have modified the description. "Positive matrix factorization shows presence of significant primary fossil fuel and biomass burning organics."

(5) L41 at-during

**Authors' reply:** done

(6) L44-45, I don't think the conclusion is from your solid result. Most you speculate these results. (7) L44-47 the conclusion cover all the possible. I would ask the author revise it carefully. What is your conclusions during the sampling period. If these solid conclusions are not from this study, you need to remove it. Seemly, I like to see what you find on BC particles not for haze formation.

**Authors' reply:** The conclusions are based on our results, but are indeed only for BC-particles and two specific cases during the sampling period. We agree that over-interpretation should be avoided. Therefore, we have carefully revised this sentence. "However, for individual pollution events, sometimes primary species could also play a dominant role, as revealed by the compositions of BC-particles in two polluted episodes during the sampling period."

(8) L56 Morphology of BC might be altered greatly. These citations don't supply any morphology of BC particles. You need find others from electron microscopies.

**Authors' reply:** We have cited a couple of electron microscopic studies, including: Wang, Y., Liu, F., He, C., Bi, L., Cheng, T., Wang, Z., Zhang, H., Zhang, X., Shi, Z., and Li, W.: Fractal Dimensions and Mixing Structures of Soot Particles during Atmospheric Processing, Environmental Science & Technology Letters, 4, 487-493, 10.1021/acs.estlett.7b00418, 2017.
Li, W., Sun, J., Xu, L., Shi, Z., Riemer, N., Sun, Y., Fu, P., Zhang, J., Lin, Y., Wang, X., Shao, L., Chen, J., Zhang, X., Wang, Z., and Wang, W.: A conceptual framework for mixing structures in individual aerosol particles, J. Geophys. Res. - Atmos., 121, 13,784-713,798, 10.1002/2016JD025252, 2016.

(9) L77, I don't agree with the claim. For example, Wu et al., 2017. Size distribution and source of black carbon aerosol in urban Beijing during winter haze episodes. Atmos. Chem. Phys. 17 (12), 7965-7975. The study seemly, give the online BC-containing particles in Beijing.

**Authors' reply:** Sorry, our claim is not clear, we meant to say that no chemical characterization of BC-containing particles only. It is now changed. And in fact, the work mentioned here was already cited as a previous work on BC-particles in Beijing.

(10) L161 Discussion, deleted s

**Authors' reply:** done

(11) L260-261, L284-285, L331-332 L347-348, all the parts discussed the aqueous reactions for nitrate and SOA formation during the nighttime. I take a look at the data from the study. It is too simply to get such conclusion. I might ask the authors cite more related references here. For example, Wu et al., 2018. Environmental Science & Technology Letters 5 (3), 160-166;Sun et al., 2018. Journal of Geophysical Research: Atmospheres 123 (2), 1234-1243.Kuang et al., 2016. Geophysical Research Letters 43 (16), 8744-8750.

**Authors' reply:** Thanks for the references provided. The aqueous-phase production of secondary species was a possible pathway in a qualitative manner. The references provided, which are also conducted in NCP, in fact strongly support our postulated aqueous-phase pathway therefore are very useful. They have now been cited along with our discussion in the main text. "Similarly, nitrate and sulfate formations driven by high RH in North China Plain have been proved previously (Kuang et al., 2016; Sun et al., 2018;Wu et al., 2018)."

(12) L278-279 I don't understand the sentence. Why was the large decrease of organics coating concentration?

**Authors' reply:** Since $R_{BC}$ is the ratio of concentrations of total coating material to BC cores. There was a drop of $R_{BC}$ at 4:00pm, however, since nitrate/BC, sulfate/BC and chloride/BC did not decrease around 4pm (there was even an increase of nitrate/BC), the decrease of $R_{BC}$ must be caused by the decrease of organic/BC. And Fig.4f further shows that it is in fact the portions of BBOA and FFOA decreased since the OOA1/BC and OOA2/BC in fact increased. We have revised the sentence in the text. "In fact, the 4:00pm $R_{BC}$ drop was mainly caused by the large decrease of Org/BC (as $SO_4^{2-}$/BC, $NO_3^-$/BC and $Cl^-$/BC did not decrease at 4:00pm, Fig. 5d) - mainly the portions of fossil fuel and biomass burning OA (Fig. 4f). "

(13) L292 at-during

**Authors' reply:** done

(14) L307 This can be expected for urban aerosols. I don't understand it. Why?

**Authors' reply:** We have expanded this sentence. "As BC-containing particle in urban Beijing were likely influenced by multiple local/regional primary sources, relative amount of secondarily formed coating species would be less than those of highly aged BC, therefore a lower $R_{BC}$ is expected."

(15) L328 of-at (16) L269 miss comma after ws (17) L317 at two polluted episodes

**Authors' reply:** done

(16) For section 3.5.2 Seemly, the authors found different coating species on soot particles. FE, the author found large SOA; SE the author proposed large POA instead of SOA. Do the authors answer how POA associated with BC? If these particles were emitted from sources, these mixing should occur in all the time, not just SE. Were there different sources in SE and FE? Seemly, the author didn't supply any wind and backtrajectories here. I would ask the authors carefully check the data. Make sure the differences in FE and SE are large. Here the authors only compared the organics. What about the sulfate and nitrate are in the coating of BC there. I am certainly struggling on the part.

**Authors' reply:** As requested, we have added the back trajectories, wind rose plots, as well as the vertical distributions of wind speeds/directions during these two episodes. It shows clearly that these episodes are very different, and therefore they would have

different sources. Yes, it is likely that a portion of the POA associated with BC might come from the sources (they mixed together with BC from the source), and such a portion might be present all the time. Nevertheless, what we try to demonstrate here is that: during FE, significant production of secondary species (likely from aqueous-phase reactions) of sulfate, nitrate and OOA1 led to the significant increase of BC-particles, while during SE, it was mainly the POA led to the extremely high loading of BC-particles. This can be seen clearly from the composition pie charts (Figs. 7a and 7b). Also, please note the major inorganic components including sulfate, nitrate, chloride and ammonium were also shown. Increases of inorganic species were also observed during FE not only organics, while during SE, inorganic species were significantly lower than those of the average case. We have modified the writing to make our arguments clear and straightforward.